# Data-Driven Smart Living Lab to Promote Participation in Rehabilitation Exercises and Sports Programs for People with Disabilities in Local Communities [note 1]

**DOI:** 10.3390/s23052761

**Published:** 2023-03-02

**Authors:** Seung Bok Lee, Yim Taek Oh, Seung Wan Yang, Jong Bae Kim

**Affiliations:** 1Yonsei Enabling Science and Technology Research Center, Seoul 26493, Republic of Korea; 2Korea Wheelchair Rugby Association, Seoul 05540, Republic of Korea; 3Frontier Research Institute for Convergence Sports Science, Yonsei University, Seoul 03722, Republic of Korea; 4Department of Occupational Therapy, College of Health Sciences, Yonsei University, Wonju 26493, Republic of Korea

**Keywords:** smart medical, data, living lab, rehabilitation sports and exercise, disability, community

## Abstract

Patients discharged from hospitals after an inpatient course of medical treatment for any ailment or traumatic injury that results in disabling conditions and are rendered mobility impaired require ongoing systematic sports and exercise programs to maintain healthy lifestyles. Under such circumstances, a rehabilitation exercise and sports center, accessible throughout local communities, is critical for promoting beneficial living and community participation for these individuals with disabilities. An innovative data-driven system equipped with state-of-the-art smart and digital equipment, set up in architecturally barrier-free infrastructures, is essential for these individuals to promote health maintenance and overcome secondary medical complications following an acute inpatient hospitalization or suboptimal rehabilitation. A federally funded collaborative research and development (R&D) program proposes to build a multi-ministerial data-driven system of exercise programs using a smart digital living lab as a platform to provide pilot services in physical education and counseling with exercise and sports programs for this patient population. We describe the social and critical aspects of rehabilitating such a population of patients by presenting a full study protocol. A modified sub-dataset of the previously generated 280-item full dataset is applied using a data-collecting system—“The Elephant”—as an example of how data acquisition will be achieved to assess the effects of lifestyle rehabilitative exercise programs for people with disabilities.

## 1. Introduction

The literature indicates that a high percentage of patients initially hospitalized for an acute ailment or injury face varying degrees of secondary complications after discharge upon completing an acute inpatient medical or rehabilitation course [1,2,3,4]. Most of these patients will have received suboptimal care at home or subacute institutions that have failed to deliver a comprehensive level of attentive care, which led to activities of dependent functioning and ultimately debilitating lifestyles [1,2,3,4]. As a result, the majority are at increased risk of hospital readmission [3,4,5], with morbidities that require highly specialized medical treatment which ultimately drive-up healthcare costs [4,6,7]. Moreover, studies in the US have substantiated cost–benefit aspects based on total annual medical and healthcare expenses for individuals with disabilities [8,9]. Researchers have suggested analyses of expenditure outcomes between individuals with disabilities who engaged in regular exercise activity as opposed to those who did not regularly participate in any fitness-type programs acutely following hospitalization [8,9].

The path leading to medical complications requiring specialized clinical care and thus readmission is commonplace [10,11]. The situation is especially true among patients who suffer from permanently disabling conditions whereby rendered paralysis—as in stroke or spinal cord injury onset—consequently results in impaired mobility [10,11] and inevitably leads to dependent lifestyles with overall poor health outcomes [11]. These patients require ongoing systematic rehabilitation programs of services available and readily accessible beyond the confines of a medical institution within their local communities. However, many communities lack the resources to deliver an encompassing program of services with appropriate content designed for individuals with varying underlying disabilities [12]. Additionally, thus, they become further isolated and demarginalized in society [13].

A collaborative multi-task assignment, overseen by four South Korean government ministries (Figure 1), seeks to address and fulfill this demarginalized population’s critical need by delegating federally funded R&D tasks to various academic and business institutions (Figure 2). This endeavor requires expertise from clinicians, allied health professionals, and specialty experts in the field of disability sports and physical education, alongside input sought from data collection and management entities. The extensive project efforts, expected to span over three years, will systematically focus on achieving three broad areas (Figure 3). The first year is dedicated to meticulously designing an innovative data-driven smart medical rehabilitation exercise and sports living lab infrastructure. The construction efforts are now nearing completion at the National Rehabilitation Center, overseen by the Ministry of Health and Welfare in South Korea. In the second year, the primary focus is developing innovative smart devices and prototype exercise equipment for installation in the living lab center. The final year will emphasize operational plans and system expansion. It will likely go beyond the projected timeframe to ensure completion of the proposed testbed with the substantiation of results from a complete dataset application to promote the nationwide dissemination of these programs delivered from state-of-the-art facilities.

Implementing and assuring the proper delivery of such services requires an architecturally barrier-free infrastructure equipped with smart digital devices and state-of-the-art exercise equipment specifically adapted for people with disabilities. A cutting-edge facility such as this is conducive to promoting participation in exercise and sports activities at an independent level. Moreover, this innovative system does not require staffing by licensed clinicians. On the contrary, it is fully operable under the supervision of non-clinical experts trained in physical education and disability sports who can provide, at best, a minimum level of coaching and guidance to facility goers.

The project’s overarching goal in the “Living Lab”—as a constituent of the R&D proposal—is to build an infrastructure that houses a data-based smart digital healthy lifestyle exercise and sports system which enables people with disabilities to actively engage in promoting healthy living and alleviate devastating medical complications after discharge from the hospital [1,3,4,5,6,7]. The detailed contents of the complete R&D project, along with the associated 280-item comprehensive database, were presented at the 2022 International Conference on Smart Health and Telematics (ICOST) held in Paris, France, and after that, described in a publication of the academic hearing [14].

This paper describes the social and critical aspects of rehabilitating such a population of patients with the full study protocol involving the application of the previously generated 280-item full dataset. The grand scheme of the planned R&D study is presented by exploiting a modified version of the full dataset using a piece of manufactured digitized data-collecting ergometer device—“The Elephant”—as an example of how data will be acquired to substantiate the long-term effects of lifestyle rehabilitative exercise among individuals with disabilities in the grand scheme of the planned study.

## 2. Methods

### 2.1. Living Lab as a Platform

Immediately after discharge from the hospital, patients will transition to a “smart medical healthy” living lab environment (Figure 4) before returning to their respective domiciles in the community. This simulated home and community environment—a data-driven smart medical living lab—serves as the platform. In this facility, all acquired data are linked with pertinent medical and health records and the personal information of individuals with disabilities acquired from medical and exercise centers throughout the nation.

Data regarding personal clinical characteristics and medical records subjectively retrieved by individuals from hospitals are acquired, entered, and stored in the data server. Information regarding current health status will be identified through a questionnaire interview. The discussion also provides the opportunity to identify personal goals and expectations, guiding an appropriate exercise regimen.

### 2.2. Prototype Device—“The Elephant”—Trial

In this testbed study, we recruited otherwise healthy able-bodied volunteers to evaluate prototype exercise equipment designed by a collaborating research team. This ergometer system is called “The Elephant” (Figure 5). It was designed to allow individuals with varying types of disabilities with an underlying compromise in mobility and functioning abilities of upper and lower limbs. Useful features of this technology include the ability to measure electromyography, electrocardiography, body temperature, and oxygen saturation levels. These variables are measured in the volunteer study participants using a sub dataset (Table 1, Table 2 and Table 3). The complete details are outlined below under “3.3 Full Study Protocol.”

### 2.3. Full Study Protocol

Herein, we describe the methods for the full study soon to be launched. The comprehensive dataset developed for the whole study includes a total of 280 data items organized under three broad categories, as follows: (1) personal information (demographics and baseline clinical characteristics), (2) evaluation data (records from medical centers), and (3) exercise data (acquired through the program of exercises using the developed equipment set up in the living lab). A subset containing 82 data items from the comprehensive dataset was used for the “Elephant Trial,” as shown in Table 1, Table 2 and Table 3.

The multi-collaborative tasks are arranged under four broad areas and are as follows: (1) systems, (2) technology, (3) data, and (4) programs. The research efforts will focus on four categories (Figure 3) which include (1) ‘partnership and collaboration,’ (2) ‘equipment and biometric technology,’ (3) ‘data linkage and integration,’ and (4) ‘exercise and physical education services.’ Each area of the research focus has been delegated and will be undertaken individually or collaboratively among the funded academic institutions and collaborating corporate and ministerial entities (Figure 2 and Figure 3). The Ministry of Science and ICT is responsible for integrating and disseminating the linked database.

The ‘Evaluation’ data consist of four categories. These include medical and community records about general health conditions, which will have been evaluated during hospitalization (dH), in follow-up clinics after discharge (iC), from personal trainers (PT) at exercise centers, and the National Insurance program of South Korea (NI). It is further sub-categorized by data acquisition from (1) body composition, (2) administrative health data (medical and surgical histories; lab tests; and diagnoses), (3) functioning status (physiology; biomechanics; and body function test), and (4) quality of life and physical assessment (social; mental; and general).

The ‘Exercise’ data consists of two categories. These include (1) biometrics of exercise (BE) using physiological sensors (e.g., heart rate and oxygen saturation during the exercise), and (2) amount of exercise (AE). The amount is assessed by performance in exercise, frequently described as FITT [15,16]; the acronym describes the parameters of frequency (# per week), intensity (exertion and speed), time (mins per one time), and type (aerobic/anaerobic, cardiac, strengthening, resistance, etc.).

The stepwise protocol in the collection, dissemination, and storage of the acquired data involves the following process: (1) initial first visit evaluation; (2) exercise regimen prescription; (3) data collection via software apps; (4) data transmission to a storage server (databank_data.3.pgm); (5) save as ‘dataset’ in the databank “data.3.pgm” under each registered participant; and (6) disseminate to ‘dataset_cloud’ and upload information to a cloud server. The stored data information can be accessed by participants, clinical providers, and all relevant individuals or entities with permission for all healthcare and research-related purposes.

#### 2.3.1. Pre-Participation Evaluation

All participants undergo a pre-participation health assessment using the ACSMs screening algorithm (Figure 6). The focus is to evaluate the current health status, underlying physical fitness, and athletic ability to determine an appropriate exercise program.

First, determine the necessary precautions before starting based on the pertinent medical records and determinants for adjusting the intensity, frequency, and amount of exercise; second, select the appropriate direction with intensity levels based on the participants’ current and expected health status along with subjectively expressed goals and expectations; and third, depending on the type of disability, define whether the underlying condition is considered static (e.g., amputee) or dynamic (e.g., cardiovascular, stroke, etc.) to determine the appropriate required level of supervision.

Data will be ascertained by extracting information with consent from prior hospitalizations and other pertinent medical records.

#### 2.3.2. Assessment of Fitness and Athletic Performance

After the initial visit and completion of the pre-participation interview, physical fitness and athletic ability are systematically assessed for appropriate prescription and program design. This focuses on ascertaining the following areas: (1) body composition, (2) cardiorespiratory fitness, (3) strength and muscular endurance, and (4) flexibility and functional movement. However, in this pilot study, we focus on physical functioning, emphasizing data from the strength assessment and manual muscle test results (Table 3).

##### Body Composition

The evaluative variables include the height, weight, body segment length, chest circumference, waist circumference, and the amount and the relative ratio of body fat (FM) and lean body mass (FFM), including the internal components. Acquired information such as the body fat percentage (PBF), fat distribution, body segment circumference, and bone density are valuable predictors of health risk factors that can promote exercise performance, leading to healthier lives for people with disability. These items are not incorporated in this testbed study of the ergometer system; however, they will be assessed as part of the comprehensive dataset in the future study.

##### Strength and Endurance

The 1-RM, 10-RM, isokinetic, and isometric tests are included for able-bodied people. Traditional methods such as 1-RM and 10-RM are evaluated using equipment conventionally available in most fitness centers; isometric and isokinetic tests require special equipment for the corresponding movement. In addition, manual muscle testing (MMT) was included to assess participants with a disability precisely. The strength and muscular endurance evaluation items are reported as data items 43 through 81 using the ergometer device for this study.

##### Range of Motion and Function

Functional movement evaluation was included to check the musculoskeletal symmetry/asymmetry and imbalanced motor function. The evaluation items for ROM and functional movement are shown in Table 2.

## 3. Results

In the previous published academic hearing [14], we presented results using a single test, namely the 6-MWT, whereby data were acquired from seven participants with underlying spinal cord injuries. In this pilot study, a different sample (*n* = 7) comprised seven non-disabled volunteers from an affiliate university. Each participant (A through G) underwent an exercise test after an interview using the prototype equipment named “The Elephant.” We acquired the preliminary data (Table 1, Table 2 and Table 3) using measurements from an ergometer device developed by the team responsible for manufacturing prototype exercise equipment.

The comprehensive study will acquire data items using other manufactured devices by collecting information regarding balance and coordination exercises and anaerobic and aerobic exercises. The data were collected using the device and survey questionnaires for this pilot run test and included 82 data items classified into 2 categories. The acquired measurements are categorized as (1) clinical examination (age, height, weight, and recent blood test results from subjectively submitted records) and (2) functioning evaluation (sub-categorized as motor strength assessed by manual muscle test and exercise equipment).

## 4. Discussion

From a rehabilitation medicine perspective, the goal is to enhance the patients’ functioning ability, thereby ensuring a safe and stable transition toward community reintegration as an essential component of recovery [18,19]. This is achieved through a medically prescribed conventional rehabilitation program by a multi-disciplinary team of medical professionals comprised of physicians and nurses along with allied health professionals, including occupational (OT) and physiotherapies (PT) [20], within a simulated home or community environment before discharge. A data-driven program focuses on rehabilitation exercises and sports activities specifically designed to aid in developing a regimen of self-directed care and health maintenance. Unlike the acute hospital course, the setting does not require qualified licensed clinicians to provide services. Thus, participating individuals are instructed by non-clinician professionals with expertise in disability sports and physical education to be safely guided throughout the critical transition toward a stable community reintegration.

### 4.1. ‘The Elephant’ Pilot Trial

The living lab serves as the platform from which the pilot services of the R&D program in rehabilitation exercise, sports, and physical education are rendered for individuals with a disability in the local community. Completing the design, construction, and subsequent operations of the living lab has been the crux of the multi-ministerial R&D project. As construction is nearing its end, the focus is shifted toward applying the generated full dataset and acquiring valuable big data. The overarching goal is to develop a system that collaboratively applies the generated big dataset by using the results of a successful operation in the testbed phase of the service delivery program. The collected data using the developed smart medical devices and exercise equipment set up within the living lab will merge with continuously received information using a cloud server.

Against the backdrop of the R&D efforts, the living lab—equipped with smart digital devices and equipment such as ‘The Elephant’—offers promising feasibility in substantiating exercise’s beneficial chronic health effects among the elderly and individuals with disabilities. Thus, applying this data-driven smart medical platform among community-dwelling individuals with various medically diagnosed disabilities will ultimately lead to a nationwide establishment of innovative and barrier-free fitness centers designed for this population.

It has been reported that new training equipment or systems must be evaluated for their efficacy to provide various individuals with evidence-based benefits on how multiple devices and technology may compare [21,22]. Thus, the ‘smart’ features of the manufactured ergometer system require fine-tuning with a clinical application to an appropriate cohort of individuals with disabilities. Periodic evaluations of devices and systems and participants’ progress provide clinical insight into the individuals’ overall health status. Furthermore, Falcone et al. (2015) state “that caloric expenditure is a standard measurement” that enables the comparison of different exercise systems for their underlying efficacy [21]. ‘The Elephant’ was designed to allow the propelling motion of all four limbs to aid in improving the limbs’ range of motion and simultaneously yield cardiovascular benefits. However, at the current juncture of its developmental stage, it is not equipped to measure caloric expenditure during activity performance. The innovative equipment will require further testing and evaluation for the underlying efficacy toward favorable outcomes concerning life-transforming exercise benefits.

### 4.2. Safe and Effective Exercises Regimen

The data information reported in this preliminary study—as a constituent of the complete R&D study planned for the near future—was acquired from applying a sub-dataset using an ergometer system called ‘The Elephant.’ This innovative exercise equipment was developed by the research team designated to the “Equipment and Biometrics Technology” discipline (Figure 2). It was designed to enhance mobility by safely increasing the ROM and motor strength and offers cardiovascular benefits through the simultaneous pedaling motion of all four limbs. In this study, healthy volunteers (A through G) performed the prescribed program of activities summarized in Table 1, Table 2 and Table 3.

Using the comprehensive dataset, our goal in the upcoming study is to substantiate the beneficial effects of exercise using ‘The Elephant’ along with other smart devices and equipment that will be set up in the living lab. We intend to apply the measured parameters (Table 1, Table 2 and Table 3) to a population of adults with acute and chronic disabilities resulting from spinal cord injury (SCI), musculoskeletal disorders, and cardiovascular diseases. Pre- and post-exercise outcome measurements of the outcome in functioning status concerning changes in the ROM, motor strength, and underlying mood levels will be compared and analyzed, as was done in this pilot trial. The rationale for these measurements stems from previous studies which have reported that an active lifestyle increases well-being [23] and cognitive functioning, decreasing risks of developing cognitive deficits and progression toward dementia [24], ultimately resulting in an enhanced quality of life [25].

According to the ACSMs Guidelines for Exercise Testing and Prescription (GETP), exercise programs are prescribed by the FITT principle: frequency, intensity, time, and type of exercise basis [15,16]. The abbreviation is further expanded by adding the variables volume (V) and progress (P); thus, all living lab exercise program elements will be prescribed applying the FITT-VP principle per the ACSMs recommendations. The trainer’s frequency of administration [26] and data acquisition details will also be included to ensure the safety of all participants in the complete study.

As observed in ‘The Elephant’ trial, evaluating flexibility and functional movement concerning the range of motion (ROM) for able-bodied individuals is generally less critical. On the contrary, for people with disabilities, degrees of ROM are used to distinctly classify varying types of disability by the International Paralympic Committee since other underlying disorders often accompany it. A short range of motion is associated with a decreased performance in activities of daily living. Hence, maintaining overall joint flexibility helps smooth motion implementation and injury prevention. On the other hand, tissue damage can result when the activity exceeds the joint ROM. Thus, our goal is to uphold safety as its utmost priority when applying these programs of exercise to the population of individuals with disabilities.

In this ergometer study, we also measured mood outcomes among the participants since exercise and fitness activities have also been associated with improving anxiety and depressive disorders [27,28,29]. Our future study using the comprehensive dataset will measure a broader range of parameters related to mood and depression, thereby upholding the theme of regular activity, which yields an enhanced quality of life [25]. Moreover, the literature reveals substantial evidence to support the beneficial effects of exercise, whose benefits are indisputable, outweighing the risks of performing these activities in most adult populations [17,30,31,32]. Although numerous studies have substantiated these health benefits based on the able-bodied adult population, Garber et al. (2011) guided professionals who counsel and prescribe individualized exercise to otherwise healthy adults of all ages, which can be applied to individuals with underlying chronic illnesses or disabilities [30].

The limitations identified from this pilot trial included the modest sample size (*n* = 7) and baseline clinical characteristics of the target population, which comprised healthy, non-disabled participants. Thus, the generalizability of the results will be compromised. The future full study plans to recruit a larger sample population (*n* = 60) of 30 participants in each of the two arms: experimental versus the control.

In summary, the multi-ministerial collaboration of data acquisition and the linkage of big data acquired using innovative technology to promote the delivery of a program of services is a feasible plan for meeting the critical healthcare needs of individuals with chronic disabilities in local communities. The integrative research and development benefits are multi-fold. We expect the beneficial effects will impact society’s medical, technological, and economic advancements.

## 5. Conclusions

The study did not yield results to reflect significant changes in the underlying health parameters measured in the healthy volunteer cohort of seven, as shown by the data acquisition. As for the prototype technology, the “The Elephant” ergometer requires fine-tuning and further evaluation for optimal applicability among the future study population of individuals with varying disabilities.

## Figures and Tables

**Figure 1 sensors-23-02761-f001:**
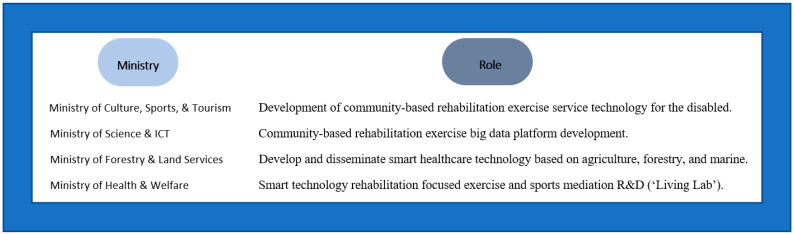
South Korean government ministries and their roles.

**Figure 2 sensors-23-02761-f002:**
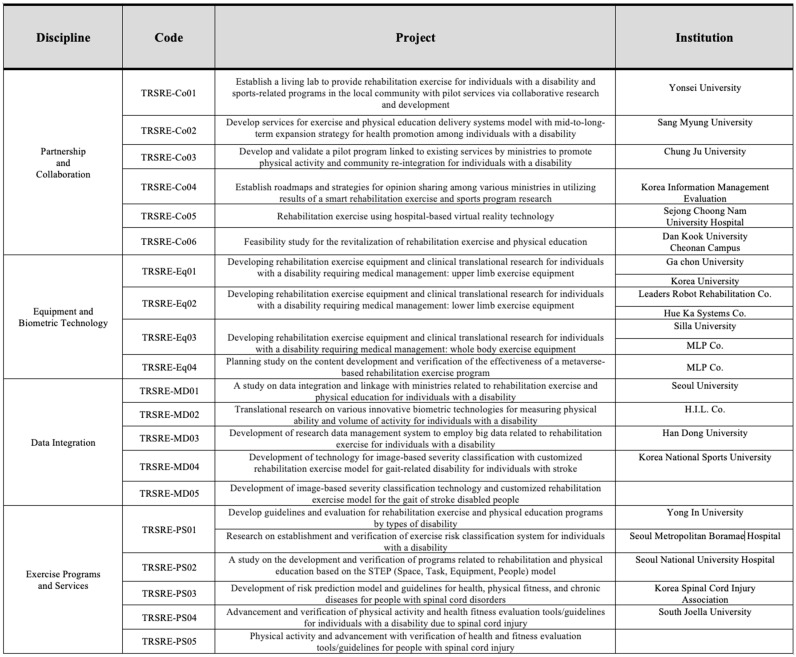
The four disciplines in rehabilitation exercise and sports translational research and development are subdivided into various projects delegated to academic and corporate institutions as part of the federally funded R&D project.

**Figure 3 sensors-23-02761-f003:**
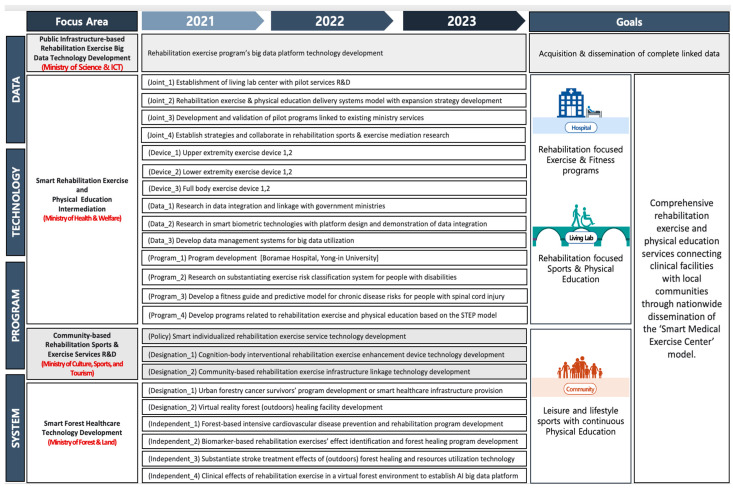
Focused research areas—system, program, technology, and data with respective projected goals—overseen by respective South Korean government ministries.

**Figure 4 sensors-23-02761-f004:**
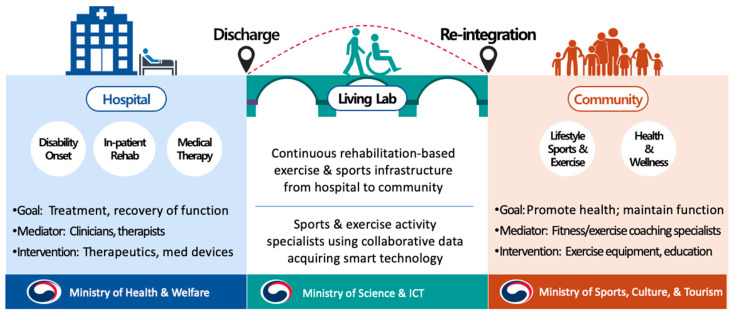
‘Living Lab’ model as a testbed providing continuous rehabilitation for community-dwelling individuals with disabilities.

**Figure 5 sensors-23-02761-f005:**
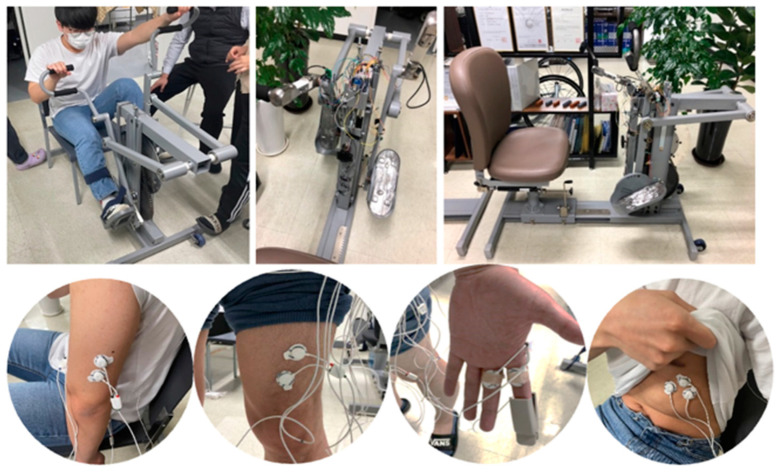
The elephant is an ergometer system developed by the collaborating research team overseeing “Equipment and Biotechnology.”.

**Figure 6 sensors-23-02761-f006:**
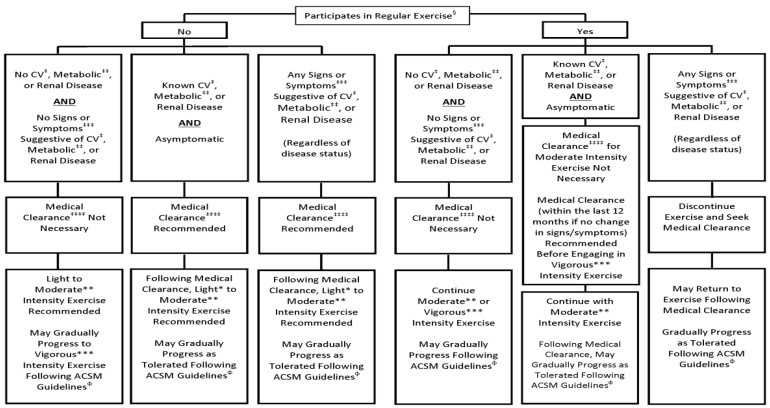
ACSMs recommendations for exercise preparation health screening [17]. §: Exercise participation, performing planned, structured physical activity at least 30 min at moderate intensity on at least 3 days per week for at least the last 3 months. *: Light-intensity exercise, 30% to G40% HRR or $V^{\;\cdot }$ O2R, 2 to G3 METs, 9–11 RPE, an intensity that causes slight increases in HR and breathing. **: Moderate-intensity exercise, 40% to G60% HRR or $V^{\;\cdot }$ O2R, 3 to G6 METs, 12–13 RPE, an intensity that causes noticeable increases in HR and breathing. ***: Vigorous-intensity exercise Q60% HRR or $V^{\;\cdot }$ O2R, Q6 METs, Q14 RPE, an intensity that causes substantial increases in HR and breathing. ‡: CVD, cardiac, peripheral vascular, or cerebrovascular disease. ‡‡: Metabolic disease, type 1 and 2 diabetes mellitus. ‡‡‡: Signs and symptoms, at rest or during activity, includes pain, discomfort in the chest, neck, jaw, arms, or other areas that may result from ischemia; shortness of breath at rest or with mild exertion; dizziness or syncope; orthopnea or paroxysmal nocturnal dyspnea; ankle edema; palpitations or tachycardia; intermittent claudication; known heart murmur; or unusual fatigue or shortness of breath with usual activities. ‡‡‡‡: Medical clearance, approval from a health care professional to engage in exercise. Φ: ACSM Guidelines, see ACSM_s Guidelines for Exercise Testing and Prescription, 9th edition, 2014.

**Table 1 sensors-23-02761-t001:** Clinical examination (basic profile and blood test).

Data	Unit	A	B	C	D	E	F	G	Mean	Std Dev	Min	Max
Age	years	22	22	24	24	22	23	23	22.9	0.9	22	24
Height	cm	158	153	175	170	166	156	169	163.9	8.2	153	175
Weight	kg	44	55	71	64	56	48	60	56.9	9.2	44	71
Na^+^		139	144	142	138	140	140	139	140.3	2.1	138	144
K^+^		3.4	3.6	4.7	4	3.5	3.5	3.4	3.7	0.5	3.4	4.7
Ca^2+^		8.4	8.7	10.3	9.5	8.5	8.5	8.4	8.9	0.7	8.4	10.3
BUN		14	10	28	24	12	12	10	15.7	7.3	10	28
Cr		0.75	0.46	1.15	1.39	0.52	0.52	0.46	0.8	0.4	0.46	1.39
Glu (Fasting)		68	74	95	97	70	70	74	78.3	12.3	68	97
WBC		4000	5300	9500	8800	4700	4700	5300	6042.9	2177.0	4000	9500
Hgb		14.7	12	18.5	17.8	13.9	13.9	12	14.7	2.6	12	18.5
Cholesterol total		187	199	213	235	190	190	199	201.9	17.0	187	235
Plt		170,000	150,000	420,000	350,000	170,000	170,000	150,000	225,714.3	111,034.1	150,000	420,000
HbA1C		5.3	3.9	6.5	6	5	5	3.9	5.1	1.0	3.9	6.5
LDL		98	110	125	130	100	100	110	110.4	12.7	98	130
HDL		63	65	51	43	64	64	65	59.3	8.7	43	65
TG		113	130	129	175	115	115	130	129.6	21.5	113	175
AST		20	19	31	29	20	20	19	22.6	5.1	19	31

**Table 2 sensors-23-02761-t002:** Functioning: range of motion (ROM).

Data	Unit	A	B	C	D	E	F	G	Mean	Std Dev	Min	Max
Neck Flexion	degrees	52	51	47	45	52	80	51	54.0	11.8	45	80
Neck Extension	degrees	53	52	49	47	52	20–30	52	50.8	2.3	47	53
(L) Shoulder Flexion	degrees	192	189	181	180	192	180	180	184.9	5.8	180	192
(R) Shoulder Flexion	degrees	193	190	181	180	192	180	180	185.1	6.2	180	193
(L)Shoulder Extension/HyperExtension	degrees	65	63	55	55	66	50	50	57.7	6.9	50	66
(R) Shoulder Extension/HyperExtension	degrees	70	60	58	55	66	50	50	58.4	7.6	50	70
(L) Shoulder Abduction	degrees	185	183	181	180	183	180	180	181.7	2.0	180	185
(R) Shoulder Abduction	degrees	185	183	182	180	183	180	180	181.9	2.0	180	185
(L) Shoulder Adduction/HyperAdduction	degrees	55	49	46	45	57	0	40	41.7	19.3	0	57
(R) Shoulder Adduction/HyperAdduction	degrees	55	50	45	45	57	0	40	41.7	19.3	0	57
(L) Elbow Flexion	degrees	142	136	132	133	140	145	142	138.6	5.0	132	145
(R) Elbow Flexion	degrees	139	135	132	133	140	145	139	137.6	4.5	132	145
(L) Elbow Extension	degrees	181	180	180	180	180	0	181	154.6	68.2	0	181
(R) Elbow Extension	degrees	181	180	180	180	180	0	181	154.6	68.2	0	181
(L) Wrist Flexion	degrees	82	76	74	74	80	90	82	79.7	5.7	74	90
(R) Wrist Flexion	degrees	80	76	74	74	80	90	80	79.1	5.5	74	90
(L) Wrist Extension	degrees	72	67	65	65	70	70	72	68.7	3.0	65	72
(R) Wrist Extension	degrees	72	67	65	65	70	70	72	68.7	3.0	65	72
(L) Hip Flexion	degrees	142	137	115	113	141	120	142	130.0	13.4	113	142
(R) Hip Flexion	degrees	143	135	115	114	141	120	143	130.1	13.3	114	143
(L) Hip Extension	degrees	28	27	25	25	29	15	28	25.3	4.8	15	29
(R) Hip Extension	degrees	27	27	25	25	29	15	27	25.0	4.6	15	29
(L) Hip Abduction	degrees	55	54	55	53	56	45	55	53.3	3.8	45	56
(R) Hip Abduction	degrees	55	54	55	53	56	45	55	53.3	3.8	45	56

**Table 3 sensors-23-02761-t003:** Functioning: strength test.

Data	Unit	A	B	C	D	E	F	G	Mean	Std Dev	Min	Max
MMT1_(L) shoulder_abduction (deltoid_C4)	Grade (1–5)	5	5	5	5	5	5	5	5.0	0.0	5	5
MMT2_(R) shoulder_abduction (deltoid_C4)	Grade (1–5)	5	5	5	5	5	5	5	5.0	0.0	5	5
MMT3_(L) elbow_flexion (biceps_C5)	Grade (1–5)	5	5	5	5	5	5	5	5.0	0.0	5	5
MMT4_(L) elbow_extension (triceps_C7)	Grade (1–5)	5	5	5	5	5	5	5	5.0	0.0	5	5
MMT5_(R) elbow_flexion (biceps_C5)	Grade (1–5)	5	5	5	5	5	5	5	5.0	0.0	5	5
MMT6_(R) elbow_extension (triceps_C7)	Grade (1–5)	5	5	5	5	5	5	5	5.0	0.0	5	5
MMT7_(L) wrist_flexion (C6)	Grade (1–5)	5	5	5	5	5	5	5	5.0	0.0	5	5
MMT8_(R) wrist_flexion (C6)	Grade (1–5)	5	5	5	5	5	5	5	5.0	0.0	5	5
MMT9_(L) finger_flexion (C8)	Grade (1–5)	5	5	5	5	5	5	5	5.0	0.0	5	5
MMT10_(R) finger_flexion (C8)	Grade (1–5)	5	5	5	5	5	5	5	5.0	0.0	5	5
MMT11_(L) finger_abduction (T1)	Grade (1–5)	5	5	5	5	5	5	5	5.0	0.0	5	5
MMT12_(R) finger_abduction (T1)	Grade (1–5)	5	5	5	5	5	5	5	5.0	0.0	5	5
MMT13_(L) hip_flexion (L2)	Grade (1–5)	5	5	5	5	5	5	5	5.0	0.0	5	5
MMT14_(R) hip_flexion (L2)	Grade (1–5)	5	5	5	5	5	5	5	5.0	0.0	5	5
MMT15_(L) hip_extension	Grade (1–5)	5	5	5	5	5	5	5	5.0	0.0	5	5
MMT16_(R) hip_extension	Grade (1–5)	5	5	5	5	5	5	5	5.0	0.0	5	5
MMT17_(L) knee_flexion	Grade (1–5)	5	5	5	5	5	5	5	5.0	0.0	5	5
MMT18_(R) knee_flexion	Grade (1–5)	5	5	5	5	5	5	5	5.0	0.0	5	5
MMT19_(L) knee_extension (L3)	Grade (1–5)	5	5	5	5	5	5	5	5.0	0.0	5	5
MMT20_(R) knee_extension (L3)	Grade (1–5)	5	5	5	5	5	5	5	5.0	0.0	5	5
MMT21_(L) long-toe extensions (L4)	Grade (1–5)	5	5	5	5	5	5	5	5.0	0.0	5	5
MMT22_(L) long-toe extensions (L4)	Grade (1–5)	5	5	5	5	5	5	5	5.0	0.0	5	5
MMT23_(L) ankle_dorsiflexion (L5)	Grade (1–5)	5	5	5	5	5	5	5	5.0	0.0	5	5
MMT24_(R) ankle_dorsiflexion (L5)	Grade (1–5)	5	5	5	5	5	5	5	5.0	0.0	5	5
MMT25_(L) ankle_plantar flexion (S1)	Grade (1–5)	5	5	5	5	5	5	5	5.0	0.0	5	5
MMT26_(R) ankle_plantar flexion (S1)	Grade (1–5)	5	5	5	5	5	5	5	5.0	0.0	5	5
FMA (Fugl-Meyer Assessment)	0–100	100	100	100	100	100	100	100	100.0	0.0	100	100
FMA (Fugl-Meyer Assessment)-UpperExtremity	~34 points	34	34	34	34	34	34	34	34.0	0.0	34	34
FMA (Fugl-Meyer Assessment)-LowerExtremity	~66 points	66	66	66	66	66	66	66	66.0	0.0	66	66
FAC (Functional Ambulatory Category)	0–5	5	5	5	5	5	5	5	5.0	0.0	5	5
Brunnstrom Stage	1–6	6	6	6	6	6	6	6	6.0	0.0	6	6
Berg Balance Scale	0–56	56	56	56	56	56	56	56	56.0	0.0	56	56
SPPB (Short Physical Performance Battery)	0–12	12	12	12	12	12	12	12	12.0	0.0	12	12
MBI (Modified Barthel Index)	0–100	100	100	100	100	100	100	100	100.0	0.0	100	100
Distance (Elephant)	Km	5.1	4.75	4.75	4.75	5	4.75	4.75	4.8	0.1	4.75	5.1
Force-Strength(%) Left Arm	%	135.5	46.9	95.4	78.1	93.4	95.8	75.2	88.6	26.9	46.9	135.5
Force-Strength(%) Right Arm	%	83.3	115.5	87.7	93.4	89.7	78.8	96.5	92.1	11.9	78.8	115.5
Force-Strength(%) Left Leg	%	32.5	51.2	75.4	86.8	74.4	-	102.3	70.4	25.0	32.5	102.3
Force-Strength(%) Right Leg	%	2.3	42.4	99.4	92.2	109.4	94.3	59.3	71.3	38.6	2.3	109.4
Calories	Kcal	226.9	269.7	319.16	238.4	291.16	310.46	219.4	267.9	40.6	219.4	319.16

## Data Availability

The data presented in this study are available at https://link.springer.com/chapter/10.1007/978-3-031-09593-1_9 (accessed on 31 October 2022).

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
