# Peer review of "Data-Driven Smart Living Lab to Promote Participation in Rehabilitation Exercises and Sports Programs for People with Disabilities in Local Communities†"

_sensors, 2023, doi:10.3390/s23052761_

Round 1

Reviewer 1 Report

The current work proposes a solution for patients with some kinds of disabilities after a period of hospitalization, based on rehabilitation therapy and sports in a living lab setup.

The authors present a prospect of the development of the project, together with a methodology for the assessment of a therapeutic approach.

The paper is written using a correct English form, yet it is not always easy to follow.

One concern is related to the clarity of the description of the whole methodology, in 3, where the authors use several lists that in my opinion need an explanation and a motivation. 

Also, the state of the development is not clear, since along the paper the authors write about what they will do, then present results of a pilot study that seems to be carried on using different methods (“based on The Elephant”).

Here are my notes, divided per sections.

1)
While the addressed issue is clearly explained, it sounds like the authors are proposing a solution based on a center (i.e. an infrastructure) or a system (what kind?). Even if the concept may be cleared later, there should be a clear statement about it in the intro.

The authors state "this study aims to promote...", that is clearly the aim of the project, while they should also underline and clarify the aim of the current research work.

The authors may also discuss several kinds of alternative technological solutions (i.e. virtual exercises). 

2)

“Entities will work toward data acquisition and service connection using various organizational resources.”
Here, and also in other following sentences, the use of the future leads to doubts on what has been done and what is a future work. Please clarify.

3) 

Same doubt as 2, is this a proposal or an implementation? The authors mentioned a prototype in 2, and here state that “the center will have a server system”. Also, for example, among others, in 3.2.1 “The above items will be assessed”. 

4)

The authors should explain what part of the whole project they are validating, and what is the meaning of the data collected

5)

While the discussion seems to be related to the whole large project initially presented, it would be helpful to have a discussion also on the rationale of the experimentation

6)

The conclusions too should include the lessons learned during the study, supported by data, while they seem to repeat an initial statement

Author Response

[Reviewer 1]

Point (1)
While the addressed issue is clearly explained, it sounds like the authors are proposing a solution based on a center (i.e. an infrastructure) or a system (what kind?). Even if the concept may be cleared later, there should be a clear statement about it in the intro.

The authors state "this study aims to promote...", that is clearly the aim of the project, while they should also underline and clarify the aim of the current research work.

The authors may also discuss several kinds of alternative technological solutions (i.e. virtual exercises). 

Response (1)

This study serves as a research proposal for a complete study planned for the near future.  The results of this pilot trial using a single piece of equipment to collect data and subsequently apply the findings to a study population comprised of individuals with varying types of disabilities, and ultimately to substantiate the effects of exercise on this population using the manufactured digital devices.  The future complete study encompasses the multimineral R&D project. The living lab serves as the platform to deliver exercises and sports education using the various manufactured innovative digital equipment, which will be set up in the living lab center.  The goal is to substantiate the effects of exercise and sports activities among the population with the varying types of disabilities we are interested in focusing on. Our previous trial run, published as an academic hearing from the ICOST 2022 conference, presented the entire dataset generated a priori, using a sub-dataset from that full dataset, and applied it to a cohort of 7 individuals with disabilities due to underlying spinal cord injuries.  And thus, the construction of the living lab center is the necessary infrastructure, and the entire program of exercise driven by data collected from the medical institutions where study participants have been previously managed will be the entire system that we are referring to through the prose.

Lastly, the suggestion regarding virtual space as a potential area for these program operations is very well taken. However, we have another study that deals with AI/VR/AR and the metaverse; thus, we will refrain from mentioning this aspect herein.

Regarding the aim, the equipment used to verify exercise effects would be categorized as secondary outcomes of this study for argument's sake. Although not the primary outcome goals related, it is to verify and understand the effect of exercise using the manufactured ergometer (“The Elephant”) as a trial run.  We expect to use the results to steer us in the right direction and eventually apply concepts to the disabled study population using the comprehensive 280-item full dataset.

The introduction prose has been edited to communicate and clarify these points as per the comment provided by the reviewer.  Thank you for your valuable input.

Point (2)

“Entities will work toward data acquisition and service connection using various organizational resources.”
Here, and also in other following sentences, the use of the future leads to doubts about what has been done and what is a future work. Please clarify.

Response (2)

Future work refers to the comprehensive study using the full 280-item dataset in operating the living lab center. In the previously published academic proceedings, a roadmap and timeline in Figure 1 to indicate the point made. In this manuscript, we also re-iterate this to show what has already been accomplished. In the first year, we achieved the designing and construction plans of the living lab infrastructure, parallel to generating the full 280-item dataset; the subsequent year was devoted to completing the actual construction of the living lab infrastructure, alongside the manufacturing efforts in creating the digital equipment and devices which will all be set up in the living lab where the operations of exercise programs will be delivered among the study participants with disabilities. Upon successfully operating the living lab as the platform, the following action plan is to execute the proposed program and substantiate the effects of this innovative data-driven exercise & sports education system among this patient population. The ultimate goal after that is to promote the dissemination of the concept and expansion of the exercise centers throughout local communities.  This has been mentioned succinctly in the introduction and the discussions.

Point (3) 

Same doubt as 2, is this a proposal or an implementation? The authors mentioned a prototype in 2, stating that “the center will have a server system”. Also, for example, among others, in 3.2.1 “The above items will be assessed”. 

Response (3)

We are proposing the system, as it is non-existent in any part of the globe; the goal is to implement all the results, including the acquired data with manufactured innovative technology setup within an infrastructure designed with barrier-free and easy accessibility among demarginalized community-dwelling individuals with chronic disabilities. The methods subsections indicate each future study participant's protocol when initiating the program at the living lab.  We have modified the methods sections per the reviewer’s comments to communicate the ideas better.

Point (4)

The authors should explain what part of the whole project they are validating, and what is the meaning of the data collected

Response (4)

In this study, we refer to the data acquired from using “The Elephant” equipment on otherwise healthy able-body participants (n=7).  The goal is to validate the effects of the exercise program prescribed for using this equipment. In turn, after analysis and further evaluation of the technology, modifications will be made and adjustments to appropriately apply them to the full study participants with varying types of underlying disabilities.  This was also discussed and edited for clearer communication of the ideas.

Point (5)

While the discussion seems to be related to the whole large project initially presented, it would be helpful to have a discussion also on the rationale of the experimentation.

Response (5)

Agreed, and thank you for pointing this out.  We have indeed discussed the rationale for using the ergometer system accordingly.  Much of this was deduced from the recommendations presented by the American College of Sports Medicine guidelines for exercise programs prescribed for able-body and individuals with disabilities.  Please refer to the discussion under the “safe and effective exercise program” subtitle.

Point (6)

The conclusions too should include the lessons learned during the study, supported by data, while they seem to repeat an initial statement.

Response 6

Thank you for this comment.  The conclusion has been edited appropriately to indicate the reviewer’s comment.  Please see the “conclusion” section.

Reviewer 2 Report

First of all, congratulations on your work. Please find the recommendations below as an opportunity to improve the manuscript.

Overall the article is well-written. However, there is a clear issue related to the extreme similarity between this and the one published in the proceedings of the "International Conference on Smart Homes and Health Telematics". A new perspective could be introduced to allow the introduction of differences between the two articles. I recommend changing the background while citing the article and study in question.

Also, there appears to be new data that is not crossed with the one from the previous article (e.g., the samples have the exact same number, but they appear to be of different individuals).

Despite understanding that the plan is to indicate the roadmap, and describe the findings to the present moment, an article that targeted specifically the roadmap and the choices included for each discipline (Figure 2) (e.g. the type of assessment, the type of device, etc.) would differentiate more clearly the two studies, whilst providing ground to a new article that would focus solely on the data collected.

Author Response

[Reviewer 2]

Point (1) Overall the article is well-written. However, there is a clear issue related to the extreme similarity between this and the one published in the proceedings of the "International Conference on Smart Homes and Health Telematics". A new perspective could be introduced to allow the introduction of differences between the two articles. I recommend changing the background while citing the article and study in question. 

Response (1)                                                                                                                                     Yes, thank you for this suggestion. The “background” has been modified, and the article itself has been cited accordingly, per the reviewer’s comment.  The initial manuscript published as proceedings from the academic meeting focused on the broad aspects of the complete study.  As you mentioned, it included very scant data collected from a sample population of seven spinal cord-injured para-athletes.  The raw data was acquired using a simple test called “6-MWT,” which was modified for applicability to wheelchair users via propulsion as opposed to ambulation. However, the second paper focused on a completely different study population with the same number of participants. Herein, the instructions from the conference general committee chair were to provide a 50% extension of the original study from that publication along with a different title for the subsequent manuscript submission.  Thus, we are following these instructions and, therefore, inevitable that the two manuscripts convey similarities.

Furthermore, these participants were able-body volunteers tested using a newly manufactured prototype exercise equipment.  Our goal is to implement this device and apply the same program for exercise activities to acquire new data from various individuals with underlying disabled conditions throughout the upcoming complete study in the final year of this three-year R&D project.  This has been re-edited in the prose of the full manuscript, accordingly.

Point (2) Also, there appears to be new data that is not crossed with the one from the previous article (e.g., the samples have the exact same number, but they appear to be of different individuals).       

Response(2):                                                                                                                                     The data shown here is purposely not crossed with the previous data presented (results of the 6-MWT among n=7 spinal cord injured(SCI) study participants) in the previous academic proceedings were generated from the first pilot trial using a modest study sample (n=7) among all who carried a diagnosis of chronic SCI of varying levels according to the international spinal injury classification of neurological impairment; in this study, we tested a newly manufactured equipment, ‘the elephant’ ergometer system, targeting able-body participants since the equipment was still undergoing evaluation for efficacy with the likelihood of undergoing adjustments & fine-tuning of technological features for appropriate application to individuals with disabilities with obvious physical limitations and functioning capabilities compared to that of their able-body counterparts.

Point (3) Despite understanding that the plan is to indicate the roadmap, and describe the findings to the present moment, an article that targeted specifically the roadmap and the choices included for each discipline (Figure 2) (e.g. the type of assessment, the type of device, etc.) would differentiate more clearly the two studies, whilst providing ground to a new article that would focus solely on the data collected.            

Response(3)                                                                                                                                      In response to this comment: this paper is an extension of the first manuscript (publication from academic hearings at ICOST 2022)—as per instructions from the conference committee chair—using a different dataset focused on able-body study participants using a prototype exercise equipment (“The Elephant,” as a subsequent pilot trial) developed by another research team handling the production of the devices and equipment to be applied in the living lab study up ahead following this trial run study.

There are a total of six manufactured smart devices and equipment, one of which is this ergometer system; all of these will play significant roles as components in the complete research study that will apply the full 280-item dataset previously generated a priori in collaboration with the research team leading the data aspects in the entire R&D project. Again, these details have been modified and included in the manuscript accordingly taking into account the reviewer’s valuable input.

Round 2

Reviewer 1 Report

The current version of the paper shows a significant improvement, and the authors responded point per point to the previous notes. 

I believe the manuscript, in the current state, still requires English proofreading and an additional effort towards a clearer research statement. 

The authors propose the validation of a prototype of an ergometer, while the intro and results are also (and mainly) related to the social benefits of a rehabilitation center, that changes the scale of the reasoning.

Author Response

<2nd Revisions>

[The current version of the paper shows a significant improvement, and the authors responded point per point to the previous notes.]

--> Thank you for providing detailed feedback; the comments were extremely helpful in making appropriate revisions.

[I believe the manuscript, in the current state, still requires English proofreading and an additional effort towards a clearer research statement.]

--> It is unclear why the reviewer repeatedly emphasizes this point. Perhaps since the originating institution (Yonsei University) is in Korea and the authors have Korean names may cause some bias and raise such a concern; this is understandable. However, please be mindful that all authors, except for one, have been educated, received doctoral degrees, and completed post-doctoral experiences either in the USA or the UK. And thus, we are all bi-lingual (some of us multi-lingual) and fluent in English (written and spoken proficiency).

As for me, I am a US citizen. I have been thoroughly educated since grade 1 throughout primary and secondary schools, followed by higher education including master's and doctoral level graduate degree programs, and successfully graduated from a United States ACGME accredited medical school, thereafter, completing post-graduate training years in the USA, spanning well over two decades. Over the past four decades, I have written countless English course papers beginning in high school (in NJ) and throughout undergraduate and graduate school attendance at reputable universities in the USA. As a physician in clinical practice and academic medicine, I have also undertaken many writing courses in medical writing. I have submitted numerous manuscripts majority of which were published in US journals over the years, and this is the first experience in which anyone has made such a suggestion. However, please be reassured that it will be proofread by a fellow American who is highly proficient in English.

[The authors propose the validation of a prototype of an ergometer, while the intro and results are also (and mainly) related to the social benefits of a rehabilitation center, that changes the scale of the reasoning.]

The validation of the prototype equipment to be used for the full study will likely result after data acquisition has been achieved. In this paper, we are describing the full study protocol and presenting the ergometer device as an example of how we plan to acquire the necessary data to assess the effects of exercise using this equipment among individuals with disabilities. 

We have made appropriate revisions concerning these points suggested by the reviewer. Please refer to the following sections: 1) Abstract (Page 1; Lines 31-35); 2) Introduction (Page 3; Lines 98-104). Furthermore, please refer to the following elaboration to help clarify any remaining concerns and/or uncertainties.

Per the reviewer's comments, it was not feasible to produce an in-depth manuscript centered on this prototype equipment that was still being created. However, it was prompted by the academic conference committee chairs (ICOST 2022) who suggested that some tangible data reporting would be highly advantageous for publication. Thus, in the previous paper, a small data collection was yielded from another preliminary testbed involving spinal cord-injured participants who use wheelchairs.

In this current manuscript, a sub-dataset was used to assess the exercise effects using the ergometer among able-body participants since the device is still undergoing modifications in its development. The device required numerous adaptations and revisions to build a technology suitable for individuals with overt disabilities or physical functioning limitations due to underlying spinal cord injury, stroke, cardiovascular, and musculoskeletal disorders. Individuals with disabilities have many limitations which preclude them from using certain features of the device in its early development stage. And thus, at this juncture, an in-depth discussion focused on the ergometer is limiting and somewhat unrealistic given the sparsity in the data acquisition.

The rationale for using the ergometer system was iterated and addressed accordingly per the reviewer's previous comments. Although the ergometer device presented is not the primary focus of this current paper, it is introduced as such to merely demonstrate how we will acquire specific data information using the generated full dataset of 280 data items in operating the living lab center under the grand scheme of the future study. This full study using the proposed methods discussed in this manuscript will get underway in the next several weeks with the enrollment of participants who meet the inclusion criteria.

Furthermore, in this paper, we have outlined the full research study protocol, which involves the application of the entire dataset comprised of 280 items of data information. The methods section in this manuscript describes the plans for this future study once the infrastructure of the living lab center has been entirely built. This lab serves as the platform wherein numerous prototype equipment will be installed. The ergometer system is presented in this current paper as an example of the many pieces of equipment used for data acquisition in the full study ahead.

We are aware that added data information will allow for a more robust detailed analysis and conclusions based on the observed results to substantiate the effects of exercise using the equipment. The required modifications of the device are still in progress. It is expected to be completed to enable enrollment of participants beginning March 1, 2023, for the complete large-scale study. And thus, from this regard, the manuscript would be considered as the full study protocol.

Reviewer 2 Report

Dear authors. Thank you very much for your attention.

I believe all the issues have been solved and clarified.

Author Response

Thank you very kindly for providing detailed feedback; the comments were very helpful.